# Ilizarov Bone Transfer for Treatment of Large Tibial Bone Defects: Clinical Results and Management of Complications

**DOI:** 10.3390/jpm12111774

**Published:** 2022-10-27

**Authors:** Zheming Cao, Yiqian Zhang, Katelyn Lipa, Liming Qing, Panfeng Wu, Juyu Tang

**Affiliations:** 1Department of Orthopedics, Xiangya Hospital, Central South University, 87 Xiangya Road, Changsha 410008, China; 2Department of Bioengineering, Swanson School of Engineering, University of Pittsburgh, Pittsburgh, PA 15260, USA

**Keywords:** Ilizarov bone transfer technique, free flap, tibia bone defect, soft tissue defect

## Abstract

Backgrounds: The purpose of this study is to present our clinical experience using the Ilizarov bone transfer technique and free-flap technique in the reconstruction of large tibial bone and soft tissue defects, including an evaluation of both the management of postoperative complications and long-term outcomes. Methods: From January 2010 to May 2020, 72 patients with tibia bone and soft tissue defects were retrospectively evaluated. Either an anterolateral thigh perforator flaps (ALTP) or latissimus dorsi musculocutaneous flaps (LD), solely or in combination, were used to cover soft tissue defects. Once the flap was stabilized, an Ilizarov external fixator was applied to the limb. Follow-up was postoperatively performed at 1, 3, 6, 9, and 12 months. Results: Postoperatively, there were two cases of total and five of partial flap necrosis, and two cases of subcutaneous ulcers, which were caused by vascular crisis, infection, and hematoma, respectively. All the patients underwent Ilizarov external fixator surgery after flap recovery. A total of 16 complications occurred, including 3 cases of simple needle tract infection (antibiotic treatment) and 13 cases of complications requiring reoperation. A correlation factor analysis revealed that the main factors affecting the healing time were the defect length and operative complications. All patients with complications treated with the vascularized iliac flap eventually healed completely. Conclusions: The Ilizarov method used together with an ALTP, LD, or a combination thereof yields good clinical results for repairing large bone and soft tissue defects of the tibia, thus reducing the incidence of amputations. However, longer treatment times may be involved, and postoperative complications can occur. The vascularized iliac flap may be a suitable choice for the treatment of postoperative complications of this type of Ilizarov bone transport.

## 1. Introduction

Open and comminuted fractures of lower extremities caused by high-energy and severe trauma, usually accompanied by severe peripheral soft tissue injury or defects, as well as bone and soft tissue necrosis and infection, are thorny problems in surgery [1]. Without options for bone and soft tissue coverage and reconstruction, amputation may be unavoidable [2]. Therefore, finding efficient reconstruction methods to salvage the limb is imperative. Surgeons must choose a limb salvaging strategy that potentially improves the patients’ quality of life based on the fundamental principle of soft tissue coverage with osseous reconstruction.

With the development of perforator flap technique, the damage to the donor site can be significantly reduced, and the application scope is becoming increasingly wide. “Free flap” is the main method to repair large soft tissue defects, and latissimus dorsi musculocutaneous (LD) and anterolateral thigh perforator (ALTP) flaps are commonly used [3,4,5]. These flaps can not only repair wounds and prevent deep tissue necrosis but also reduce the incidence of local infection and enhance patient comfort. The blood supply of transplanted tissue provides a good microenvironment for fracture healing. Therefore, in the treatment of composite bone and soft tissue defects, we tend to use the free-flap technique to repair the wound after debridement. However, with the larger flap area required for severely injured wounds, the risk of necrosis and other complications cannot be ignored. To reduce the corresponding risk, it is also necessary to conduct a thorough debridement of the wound cover as early as possible.

At present, there are a variety of repair methods for bone defects, including allografts [6], cancellous bone autografts [7], pedicled vascularized iliac crest grafts [8], pedicled vascularized fibular transfers [9], and microvascular fibular transfers [10]. Ilizarov’s distraction osteogenesis method has proved particularly effective in large bone defect reconstruction [11], whereas other repair methods are limited by the length of the bone defect [9,10,12]. Related studies have combined free-flap transplantation and Ilizarov bone transport to achieve the reconstruction of tibia and surrounding soft tissue defects, resulting in this technique becoming popular in the field of limb repair and reconstruction [13,14,15]. However, difficulties and complications associated with the use of the Ilizarov bone transport technique have been reported [16,17,18]. These complications remain clinically intractable, prolonging treatment time and increasing patient suffering.

In this study, a retrospective study was conducted to collect the data of patients with severe large tibial bone defects combined with soft tissue defects who were treated in our department from January 2010 to May 2020. Specifically, the purpose of this report was to evaluate the clinical efficacy of Ilizarov bone transport combined with free-flap transplantation, the management of postoperative complications, and long-term outcomes.

## 2. Materials and Methods

### 2.1. Patients

Seventy-two patients with lower extremity tibia and soft tissue defects were treated using either an ALTP, LD, or a combination thereof to cover soft tissue defects. Ilizarov bone transport was used to reconstruct bone defects from January 2010 to May 2020. A total of 63 male patients and 9 female patients were included in this study. The age of patients ranged from 8 to 57 years (median, 29.5 years). The original mechanisms of injury were traffic accidents (*n* = 54), crush injury (*n* = 11), and agricultural accidents (*n* = 7). Patient details are shown in Table 1, Table 2 and Table 3. This study followed the ethical guidelines of the Hospital Ethical Committee of Xiangya Hospital, China.

Of the included patients, 28 were admitted to the emergency department after direct trauma and 44 had a history of surgical treatment, including an open reduction with internal and external fixation before admission. After debridement, the size of the soft tissue defects ranged from 50.0 to 360.0 cm^2^ (median, 198.8 cm^2^), and the size of the bone defect ranged from 5 to 18 cm (median, 10.6 cm).

### 2.2. Surgical Technique

After the initial debridement, all patients were treated by general external fixation. Flap coverage was performed when the wound was deemed clean. Massive tibial bone defects were usually accompanied by large soft tissue defects, and thus, an ALTP, LD, or a combination thereof were used to cover large defects [3].

After free-flap stabilization (median, 1.5 months), the temporary external fixator was replaced by an Ilizarov external fixator [19,20]. The distance between the fracture site and adjacent wires or half-pins was approximately 5 cm. The wires or half-pins near the knee and ankle were parallel to the joint line. Necrotic bone was excised until the boney ends presented punctuated cortical bleeding. Alternatively, percutaneous osteotomy was performed at the healthy metaphysis of the tibia, and bone transport was carried out.

The patients underwent supervised daily physiotherapy, including active and passive range-of-motion exercises for the knee and ankle. Standing and walking, with a gradual shift from partial to full weight-bearing, was encouraged. Distraction commenced on postoperative day 7 at a rate of 1 mm/d and continued in four equal increments until the desired length had been achieved. Consolidation was considered sufficient when the formation of a bridging callus was visible on three of four visible cortices in the anteroposterior and lateral radiographs, and when tenderness at the fracture site and pain during full weight-bearing without connecting rods were no longer present. After confirming consolidation, the apparatus was removed under local anesthesia in the outpatient clinic.

The postoperative complications of Ilizarov bone transport included fracture nonunion, intolerability, and soft tissue interposition (when reoperation was required to remove ischemic bone and/or embedded soft tissue before vascularized iliac bone flap transplantation) in addition to refracture (with a cancellous bone graft of iliac crest).

Demographic and intraoperative data, early complications, and long-term follow-up results were collected. Follow-up was performed at 1, 3, 6, 9, and 12 months postoperatively. Confirmation of bone healing (removal of external fixator) was monitored for up to 1 year. The cosmetic appearances of the recipient and donor sites were subjectively evaluated by the guardian and objectively by a blinded third-party observer according to a scale of 1 (excellent) to 4 (poor). The bone and functional results were assessed by ASAMI criteria [21] at the last clinical visit.

### 2.3. Statistical Analysis

Descriptive statistics of preoperative, intraoperative, and postoperative basic conditions were performed in this study. Quantitative data were expressed as mean ± standard deviation, and qualitative data were expressed as the number of cases. Student’s *t* test (a comparison of two independent samples) or a one-way ANOVA-LSD test (a comparison of three independent samples) were used to compare the factors related to fracture healing time. Statistical analysis was performed by SPSS 20.0 software (IBM Corp, Armonk, NY, USA). A p-value of *p* < 0.05 was considered to indicate statistically significant differences.

## 3. Results

The follow-up time ranged from 25 to 49 months (median, 36 months). The mean soft tissue defect size was 198.8 ± 50.8 cm^2^. To cover these soft tissue defects, ALTP flaps were used in 24 patients, LD flaps in 32 patients, and combined flaps (bilateral ALTP or ALTP + LD) in 16 patients. The flap harvesting time was 74.0 ± 20.2 min, and the total operation time was 275.8 ± 55.1 min. There were two cases of total flap necrosis, five cases of partial flap necrosis, and two cases of subcutaneous ulcers, caused by vascular crisis, infection, and hematoma, respectively. Following vascular exploration, re-flap surgery, and dressing change, all flaps successfully healed. There was a delayed healing of the donor site in two patients in which healing occurred after dressing change.

All patients underwent Ilizarov external fixator surgery after the flap was stabilized (mean, 1.5 months). The operation time was 60.6 ± 8.4 min, the external fixator was carried for 21.3 ± 5.6 months, the external fixation index was 62.4 ± 11.2 d/cm, and the bone healing time was 23.3 ± 5.6 months. Based on the ASAMI scoring, excellent and good ratings of bone results were achieved in 90.3% (65/72) of cases, and a functional result in 70.8% (51/72). Satisfaction with the cosmetic appearance of patients’ lower extremities was evaluated by their guardians and third-party blind observers, and the rates for excellent and good results were 70.8% and 68.1%, respectively.

During the follow-up period, 16 complications occurred, including 3 cases of simple needle tract infection (antibiotic treatment) and 13 cases of complications requiring reoperation. A total of 21 patients could not tolerate the external fixator because of the long treatment time and the severity of complications. The detailed management of complications requiring reoperation is shown in Table 4 and all patients eventually healed completely.

A univariate analysis showed that the time of bone healing was related to age, bone defect length, size of the soft tissue defect, and postoperative complications (*p* < 0.05, Table 3). However, the gender, weight, and smoking and drinking history of patients had no significant effect on healing time. After adjusting for confounding factors, it was found that the main factors affecting healing time were the defect length and whether complications occurred. The length of the bone defect was positively correlated with the area of the defect and complications (*p* < 0.05, Table 5). The healing time of the group with complications was longer than that of the group without complications (*p* < 0.05, Table 5).

## 4. Case Report

Case 1: A 38-year-old male presented with an open tibia and fibula fracture due to a traffic injury. After debridement, the soft tissue defects of the left leg were 168 cm^2^, and the tibia defect was 18 cm. In the first stage, an LD was used to repair soft tissue defects. After 1.5 months, the patient was readmitted for secondary Ilizarov bone transport. Distraction commenced on post-operative day 7 at a rate of 1 mm/d. After 24 months, the external fixator was dismantled, and the tibia bone was completely repaired. The ASAMI bone and function scores were good. The total time for fracture bridging and bone healing was 26 months (Figure 1).

Case 2: A 20-year-old female presented with an open tibia and fibula fracture due to a traffic injury. After debridement, the soft tissue defects of the right leg were 360 cm^2^, and the tibia defect was 14.5 cm. In the first stage, a modified LD flap was used to repair soft tissue defects. One month later, the patient was readmitted for secondary Ilizarov bone transport. Distraction commenced on post-operative day 7 at a rate of 1 mm/d. After 26 months, the external fixator was dismantled, and the tibia bone had completely healed. The ASAMI bone and function scores were good. The total healing time was 28 months (Figure 2).

Case 3: An 8-year-old male presented with an open tibia and fibula fracture due to a traffic injury. After debridement, the soft tissue defects of the right leg were 172 cm^2^, and the tibia defect was 7 cm. In the first stage, an ALTP flap and skin grafts were used to repair soft tissue defects. Two months later, the patient was readmitted for secondary Ilizarov bone transport. Distraction commenced on post-operative day 7 at a rate of 1 mm/d. After 16 months of scar hyperplasia contractures, nonunion of fractures, and intolerability, the external fixator was removed, and the repair was conducted with iliac bone and LD flaps. The total healing time was 22 months (Figure 3).

## 5. Discussion

Large-area soft tissue defects of lower extremities combined with tibial bone defects caused by trauma are a difficult surgical problem reported in numerous related studies [22]. For extremely severe bone and soft tissue defects, the patients who withstood amputation had similar or even better outcomes compared with patients who withstood reconstruction, because the former had similar sociodemographic characteristics [2] and returned to society earlier [23]. However, due to the demanding requests of most patients, reconstructing bone and soft tissue defects is challenging for surgeons. In principle, the reconstruction of this kind of injury involves using free-flap technique to first repair the soft tissue defect, and then Ilizarov bone transfer to treat the bone defect after the skin flap has completely healed and is stabilized. There are many types of free flaps available for repairing soft tissue defects, but the selection of flaps is generally based on the wound surface and the patient’s body condition [24,25,26]. Since all patients included in this study suffered from large traumatic tibial bone defects accompanied by large surface defects, LD and ALTP were the most suitable types of free flaps.

Free-flap transplantation is currently the method of choice for repairing large-area soft tissue defects. A massive flap is usually required to repair large recipient site defects, which often increases the risk of postoperative complications [27,28]. In order to avoid these complications, it is necessary to improve microsurgical techniques [29]. In this study, ALTP and LD were used to repair wounds; postoperative complications occurred in nine cases (total necrosis in two cases, partial necrosis in five cases, and subcutaneous ulcers in two cases). The following reasons for postoperative flap complications were considered: the perforator vessels could not provide sufficient supply to the whole flap covering the soft tissue defects, and a severe extravasation of blood after multiple debridement procedures of the infected wound caused subcutaneous hematoma compression of the perforator vessels. After the corresponding treatment, all patients had complete wound healing and obtained satisfactory cosmetic outcomes. Therefore, we can conclude that ALTP and LD have good efficacy in the treatment of large soft tissue defects of the lower extremities.

When the perforator flap had completely healed, the Ilizarov bone transfer technique was used to repair the bone defect. According to the ASAMI score, the bone healing was rated excellent and good in 90.3% of our patients, and the functional rating was excellent and good in 70.8%, which is consistent with the relevant results of others [30]. According to published studies of single-level Ilizarov bone transport, the average external fixation index is 2 months/cm of gained length [31,32]. In our study, the foreign fixation index was 62.4 ± 11.2 days/cm, which was comparable with the data of others. During the follow-up, 16 complications occurred, including 3 cases of simple needle tract infection (antibiotic treatment) and 13 cases of complications requiring reoperation. Considering that patients were supposed to wear the external fixator device for a considerable amount of time, complications such as soft tissue incarceration, infection, and pin loosening were unavoidable. Though it has been theorized that delayed union can be regarded as a planned staged procedure instead of a complication [33], it is our opinion that the unanticipated goals in delayed union should be classified as complications. The Ilizarov bone transfer technique involves a nonunion caused by the loss of osteogenic activity of bone tissue at the end of the bone defect after prolonged traction. Other studies have shown that some complications caused by Ilizarov bone transfer technique in the treatment of large bone defects are inevitable [18,34].

The correlation factor analysis showed that the main factors affecting healing time were the size of the tissue defect and whether complications occurred; moreover, the length of the fracture gap was positively correlated with both the area of the soft tissue defect and complications. The healing time in our study was longer than that in other relevant studies [34,35], which might be due to various factors. First, the patient’s adjustment time was relatively long due to pain in the bone slip process after surgery and the increased risk of complications due to improper surgical operation. Second, the soft tissue defect area and bone defect length of the patients in our study were larger than those of other related studies [34,35]. Since the rates of new bone formation and healing are related to the microenvironment, the healing time was still affected despite the use of flap technique for wound repair.

Based on our reported cases, the main advantages of Ilizarov bone transfer technique can be summarized as follows [34,35]: (1) The Ilizarov annular external fixator is mounted in a three-dimensional space through multiple Kirschner wires in different directions and planes, whereby the apparatus is stable. (2) The external fixator can rectify deformities such as varus and valgus and internal and external rotation in addition to correct angulation, reducing the need for secondary surgical adjustments. (3) It decreases the risk of additional vessel injuries and reduces the incidence of infection, and it can reset fracture fragments and repair almost any bone defect length. (4) It does not require bone grafting, and patients can exercise early during recovery, reducing the incidence of bedridden complications. However, Ilizarov bone transfer technique also possesses several disadvantages: (1) The technique has relatively few indications because it requires healthy metaphysis to perform the bone lengthening. (2) The circular external fixator is bulky, and the treatment process is relatively long, which often results in poor patient tolerance. (3) In the process of diagnosis and treatment, patients require numerous X-ray examinations, which increases radiation exposure. (4) Improper adjustments can cause needle tract infection and other postoperative complications. (5) In our study, the shape of the patient’s lower leg was different than that of the healthy side, and the calf had the same perimeter as the initial position of osteotomy; as a result, the patients’ satisfaction was not high. Especially in larger defects, leg lengthening occurs from the proximal tibia, resulting in the whole calf becoming as thick as the proximal tibia, often with degraded aesthetics. For patients with a primary extension, this problem would not occur because the contour of the lower leg would be maintained.

To decrease the treatment period, Yang et al. [36] and Borzunov et al. [37] modified Ilizarov bone transfer technique using double-level defect fragment bone transport and successfully decreased the bone transport time. With the development of microsurgical techniques, improving the clinical efficacy of free vascularized iliac bone flaps for the reconstruction of the mandible, tibia defects and femoral head necrosis have been demonstrated [38,39,40,41]. However, the most important limitation is the length of resection. The iliac bone flap is preferred over the fibula flap, which is commonly used to repair bone defects, because it contains both cortical and cancellous bone. Cortical bone can provide support, and cancellous bone provides a more conductive microenvironment for osteogenesis, which allows for better and faster bone healing compared with fibular flaps. In addition, the iliac bone flap has an abundant blood supply and a strong anti-infection ability. In order to solve the fracture nonunion caused by complications, repairs were conducted with free iliac bone flaps in 11 patients in this study, with healing occurring within an average of 5 months after the operation. In our opinion, the Ilizarov bone transfer technique is currently the best choice for repairing large bone defects. Nevertheless, the long carrying time and the relatively high complication rate of external fixations are clinical problems still to be resolved. Therefore, our research center has begun to gradually combine the two techniques in an attempt to solve the problem of large bone defects by reducing bone healing time and the incidence of complications. However, the specific advantages and disadvantages need to be further evaluated and defined by clinical studies on large numbers of patients.

## 6. Conclusions

The Ilizarov method combined with free flap technique yielded good clinical results in repairing large bone and soft tissue defects of the tibia, thus reducing the incidence of amputations. However, there are still the following shortcomings: (1) Compared with simple soft tissue defects, the defect area and wound severity in this study were more serious, resulting in a relatively increased risk of postoperative flap complications. (2) Since the length of bone healing time was closely related to the length of the bone defect and the occurrence of complications, the longer the treatment time, the higher the risk of postoperative complications, which cannot be tolerated by most patients. In order to solve the corresponding complications, we used the vascularized iliac bone flap in this study and achieved good results. Therefore, our center has begun to gradually explore the combination of the Ilizarov bone transport technique with vascularized iliac bone flap transplantation to reduce bone healing time and reduce the incidence of complications; however, its efficacy needs to be further observed.

## Figures and Tables

**Figure 1 jpm-12-01774-f001:**
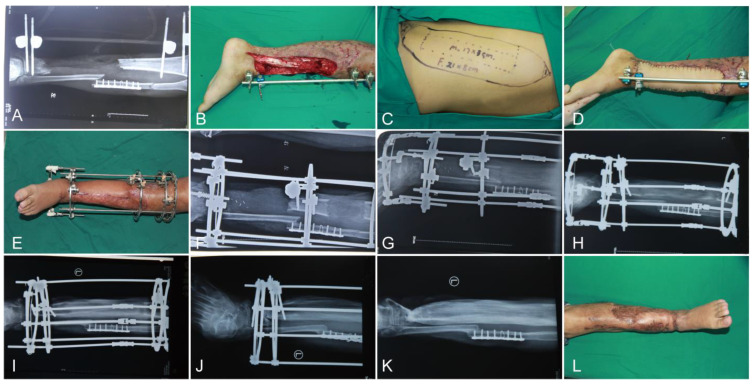
A 38-year-old male patient with right tibial bone and soft tissue defects suffered from vascular crisis. (**A**,**B**) The preoperative radiographic view and the left leg after radical debridement. (**C**) The designed latissimus dorsi musculocutaneous flap of donor site. (**D**) The soft tissue coverage of recipient site. (**E**) Installed Ilizarov external fixator at one-month follow-up. (**F**–**J**) Postoperative radiographic view at the 3-, 6-, 9-, 12-, and 24-month follow-ups. (**K**,**L**) Postoperative radiographic view and recipient site at the 26-month follow-up.

**Figure 2 jpm-12-01774-f002:**
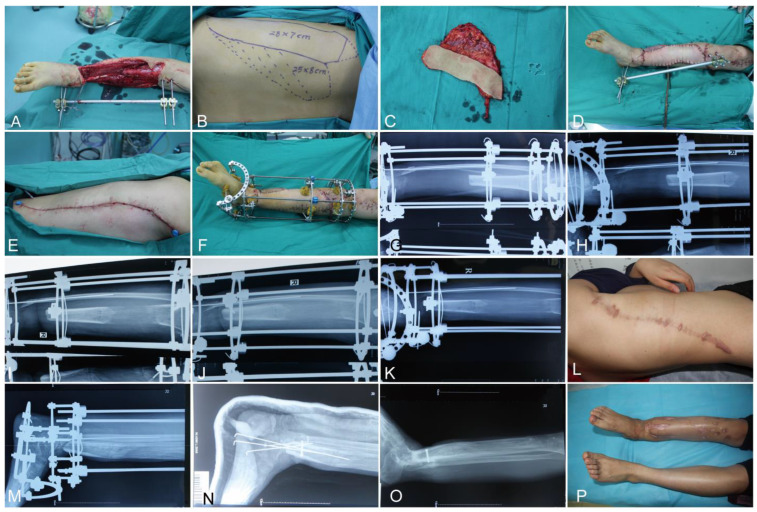
A 20-year-old female patient sustained a tibial bone and soft tissue defect on the right lower leg caused by a traffic accident. (**A**) The right leg after radical debridement. (**B**,**C**) The designed modified latissimus dorsi musculocutaneous flap of donor site. (**D**,**E**) The soft tissue of recipient and donor sites closed. (**F**) Installed Ilizarov external fixator after the soft tissue healed. (**G**–**J**) Postoperative radiographic view at the 7-day, 3-month, 6-month, and 9-month follow-ups. (**K**,**L**) Postoperative radiographic view and donor site at the 12-month follow-up. (**M**,**N**) Postoperative radiographic view before and after dismantling the external fixator at the 26-month follow-up. (**O**,**P**) Postoperative radiographic view and recipient site at the 30-month follow-up.

**Figure 3 jpm-12-01774-f003:**
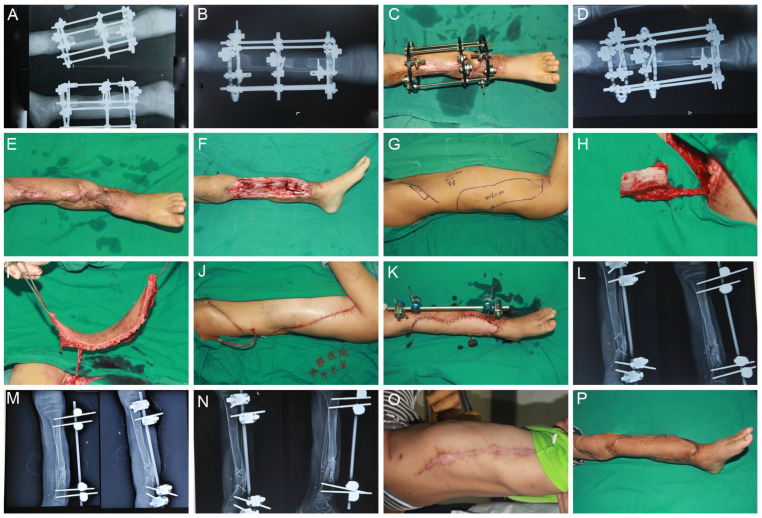
An 8-year-old male patient sustained a tibial bone and soft tissue defect on the right lower leg caused by a traffic accident. (**A**,**B**,**D**) Postoperative radiographic view at 7 days, 2 months, and 3 months after Ilizarov bone transfer. (**C**) Appearance of the leg 3 months after bone transfer. (**E**–**K**) After scar treatment, the iliac bone and LD flaps were removed to repair the defect wound. (**L**–**N**) Postoperative radiographic view at 1, 2, and 4 months after iliac bone flap surgery. (**O**,**P**) One month after removal of external fixation.

**Table 1 jpm-12-01774-t001:** Intraoperative data and short- and long-term follow-up results of soft tissue repair surgery.

Variable	Ilizarov Group (*N* = 72)
Defect size (cm^2^)	198.8 ± 50.8
Flap size (cm^2^)	209.4 ± 52.9
Soft tissue repair	
ALTP	24
LD	32
Other *	16
Flap harvested time (min)	74.0 ± 20.2
Operative time (min)	275.8 ± 55.1
Flap complications	
Total flap necrosis Partial flap necrosis Subcutaneous ulcer	2 5 2
Factors of flap necrosis	
Vascular crisis	5
Infection	2
Hematoma	2
Donor site morbidity	
Delayed wound healing	2

* Combination (bilateral ALTP or ALTP + LD) to repair the wound.

**Table 2 jpm-12-01774-t002:** Intraoperative data, short- and long-term follow-up results of bone defect repair.

Variable	Ilizarov Group (*N* = 72)
Operative time (min)	60.6 ± 8.4
Bone defect length (cm)	10.6 ± 3.3
External fixator carrying time (months)	21.3 ± 5.6
External fixation index (d/cm)	62.4 ± 11.2
Bone healing time (months)	23.3 ± 5.6
No longer tolerable	21
Deep pin tract infection	3
Reoperation	13
ASAMI functional results	
Excellent	30
Good	21
Fair	14
Poor	7
ASAMI bone results	
Excellent	46
Good	19
Fair	2
Poor	5
Cosmetic evaluation	
Subjectively ^a^	
Excellent	20
Good	31
Moderate	16
Poor	5
Objectively ^b^	
Excellent	19
Good	30
Moderate	17
Poor	6

^a^ Guardians of the patients; ^b^ blinded third-party observer.

**Table 3 jpm-12-01774-t003:** Correlation analysis of factors affecting the time of bone healing.

Variable	No. of Patients (*N* = 72)	Bone Defect Length (cm)/*p*-Value	Bone Healing Time (Months)/*p*-Value
Age (years)		0.001	0.001
<18	13	7.5 ± 1.8	17.7 ± 3.7
≥18	59	11.3 ± 3.1	24.5 ± 5.1
Age (years)		-	0.025
<18	13	(5–10)	17.7 ± 3.7
≥18	26	(5–10)	20.5 ± 3.5
Gender		0.225	0.092
Male	63	10.8 ± 3.0	23.7 ± 5.2
Female	9	9.3 ± 4.7	20.3 ± 7.3
BMI		0.722	0.518
Normal	44	10.5 ± 3.1	22.9 ± 5.2
Abnormal	28	10.8 ± 3.5	23.8 ± 6.1
Smoking history		0.447	0.194
No	44	10.3 ± 3.6	22.6 ± 5.4
Yes	28	10.9 ± 2.7	24.3 ± 5.7
Drinking history		0.606	0.273
No	49	10.4 ± 3.6	22.8 ± 5.4
Yes	23	10.9 ± 2.6	24.3 ± 6.5
Bone defect length (cm)		0.001	0.001
4–8	18	6.7 ± 1.1	16.7 ± 2.9
>8–12	29	9.7 ± 1.1	22.8 ± 2.6
>12	25	14.4 ± 1.5	28.5 ± 4.1
Soft tissue defect size (cm^2^)		0.001	0.001
<180	25	7.8 ± 2.0	19.6 ± 5.0
≥180	47	12.0 ± 2.9	25.2 ± 4.9
Postoperative complications		0.030	0.001
No	59	10.2 ± 3.1	21.8 ± 4.4
Yes ^&^	13	12.3 ± 3.7	29.6 ± 5.8

BMI, body mass index; ^&^ Deoperation is required for complications (scar hyperplasia contractures, nonunion of fracture, soft tissue incarceration, nonunion of fracture, refracture pin loosening, and intolerability).

**Table 4 jpm-12-01774-t004:** Details of patients and complications after Ilizarov bone graft requiring further surgical management.

No	Age/Sex	Cause of Injury	Bone Defect Location/Length (cm)	Wound Coverage	Initial Treatment Time (Months)	Reasons for Reoperation	Treatment Measures	Bone Healing Time (Months)	Total Disease Duration (Months)
1	8/M	Traffic accident	Left leg/7	ALTP (184 cm^2^) + Skin grafts	18	Scar hyperplasia contractures, nonunion of fracture, and intolerability	LD (175 cm^2^) + VIBF (5 cm)	4	22
2	38/M	Traffic accident	Right leg/10	LD (172 cm^2^)	23	Soft tissue incarceration and intolerability	ALTP (161.5 cm^2^) + VIBF (5 cm)	6	29
3	51/M	Crush injury	Right leg/9	ALTP (162.5 cm^2^)	23	Nonunion of fracture and intolerability	VIBF (5 cm)	4	27
4	56/M	Traffic accident	Left leg/5	ALTP (56 cm^2^)	18	Nonunion of fracture	SCIP (24 cm^2^) + VIBF (3 cm)	7	23
5	34/F	Traffic accident	Right leg/15	LD (180 cm^2^) + ALTP (126 cm^2^)	31	Soft tissue incarceration, pin loosening, and intolerability	Replace the nail, DIEP (60 cm^2^) + VIBF (4 cm)	6	37
6	42/M	Traffic accident	Left leg/13.5	LD (196 cm^2^)	31	Soft tissue incarceration and intolerability	VIBF (5 cm)	5	36
7	28/M	Traffic accident	Right leg/14	LD (160 cm^2^) + ALTP (133 cm^2^)	30	Soft tissue incarceration and intolerability	VIBF (5 cm)	5	35
8	57/M	Agricultural accident	Right leg/13	LD (161 cm^2^)	25	Nonunion of fracture and intolerability	VIBF (5 cm)	7	32
9	20/M	Traffic accident	Left leg/18	LD (189 cm^2^) + ALTP (184 cm^2^)	30	Deep pin tract infection, pin loosening, nonunion of fracture, and intolerability	Replace the nail, VIBF (5 cm)	4	34
10	27/M	Traffic accident	Left leg/15	LD (225 cm^2^)	20	Intolerability	VIBF (4 cm)	5	25
11	16/M	Traffic accident	Left leg/11	ALTP (192 cm^2^)	18	Intolerability	VIBF (5 cm)	4	22
12	47/M	Traffic accident	Left leg/16	LD (152 cm^2^) + ALTP (126 cm^2^)	28	Refracture	Cancellous bone graft of the iliac crest	9	37
13	31/M	Crush injury	Right leg/14	LD (208 cm^2^) + Skin grafts	-	Deep pin tract infection and pin loosening	Replace the nail	-	26
14	18/M	Traffic accident	Right leg/12	LD (204 cm^2^)	-	Deep pin tract infection	-	-	23
15	23/F	Traffic accident	Left leg/15.5	LD (218.5 cm^2^) + ALTP (147 cm^2^)	-	Deep pin tract infection	-	-	27
16	26/M	Traffic accident	Right leg/12.5	ALTP (212.5 cm^2^)	-	Deep pin tract infection	-	-	25

ALTP: anterolateral thigh perforator flap; TDAP: thoracodorsal artery perforator flap; LD: latissimus dorsi myocutaneous flap; PAP: peroneal artery perforator flap; DIEP: deep inferior epigastric perforator flap.

**Table 5 jpm-12-01774-t005:** The influence of related variables in the complication and non-complication groups.

Variable	Soft Tissue Defect Size (cm^2^)	Complication Group ^&^ (*N* = 13)	Non-Complication Group (*N* = 59)	*p*-Value
Bone defect length (cm) 4–8 >8–12 >12	0.001 (150.8 ± 29.9) (193.1 ± 24.2) (239.8 ± 52.8)	0.001 2/(22.5 ± 0.7) 3/(26.0 ± 3.6) 8/(32.8 ± 4.8) 13/(29.6 ± 5.8)	0.001 16/(16.0 ± 2.1) 26/(22.4 ± 2.3) 17/(26.5 ± 1.3) 59/(21.8 ± 4.4)	0.001 0.022 0.007 0.001

**^&^** Reoperation is required for complications.

## Data Availability

The data sets to support the findings of this study are included within the article, including figures and tables. Any other data used to support the findings of this study are available from the corresponding authors upon request.

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
