# Peer review of "Ilizarov Bone Transfer for Treatment of Large Tibial Bone Defects: Clinical Results and Management of Complications"

_jpm, 2022, doi:10.3390/jpm12111774_

Round 1
Reviewer 1 Report
Well designed and well presented retrospective clinical study on a high number of patients with similar injuries and largely successful limb salvage procedures, using the Ilizarov bone transport technique; I have a few comments and suggestions for textual improvement, which can be found in an attached Word file; especially in the Discussion some sections need clarifications and some re-phrasing. I have done some English language editing and hope that I did not alter any meaning. Good work ! especially in the operating room but also on paper.

Author Response
Hi Professor,
Thank you for your comments and your language editing is very helpful, we have changed the improper words according to your modification. Besides, we submitted our manuscript to MDPI for further English editing. The attachment is the edited version.

Reviewer 2 Report
Congratulations on the surgical prowess.
The article is eminently publishable (see above comments). I addresses an important clinical issue - survivability/function of mangled lower extremities and describes very well its challenges (the main value of the publications in my opinion).
Couple of points:
I did not readily see the influence of smoking status on the resuts.
Ditto influence of age.
Was early amputation (especially if transtibial amputation was possible) discussed as an option? Despite the obvious drawbacks it does offer the advantage of shorter treatment time, earlier return to function, potential less financial cost (earlier return to work, etc).
The subject has a narrow audience (for which, of course the interest is higher than the general readership)
Author Response
Hi Professor,
Thank you for your comments.
(1) Smoking history didn’t affect the bone healing time, because the P value is 0.194 (>0.05). So, we added this “However, the gender, weight, and smoking and drinking history of patient had no significant effect on healing time.” To conclude all the factors we have analyzed.
(2) Under the condition of the same bone defect length(5-10cm), the younger patients have healed faster, the patients <18 bone healing time is 17.7 months, but the patients ≥18 bone healing time is 20.5 months. To prevent readers confusion, we make the title of each factor in bold.
(3) We are grateful about your suggestion of discussing the early amputation, therefore, we added the related content in discussion: “For extremely severe bone and soft tissue defects, the patients withstood amputation had similar or even better outcomes compared to patients withstood reconstruction, because the patients withstood amputation had the similar sociodemographic characteristics(1) and return to society earlier(2) than patients withstood reconstruction.”
1.Bosse M.J., MacKenzie E.J., Kellam J.F., Burgess A.R., Webb L.X., Swiontkowski M.F., Sanders R.W., Jones A.L., McAndrew M.P., Patterson B.M., McCarthy M.L., Travison T.G., Castillo R.C. An analysis of outcomes of reconstruction or amputation after leg-threatening injuries. N Engl J Med. 2002;347(24):1924-31. doi: 10.1056/NEJMoa012604.
2.van Dongen T.T., Huizinga E.P., de Kruijff L.G., van der Krans A.C., Hoogendoorn J.M., Leenen L.P., Hoencamp R. Amputation: Not a failure for severe lower extremity combat injury. Injury. 2017;48(2):371-77. doi: 10.1016/j.injury.2016.12.001.
(4) Besides, we submitted our manuscript to MDPI for further English editing. The attachment is the edited version.

Reviewer 3 Report
The article approaches an interesting topic taking in account the importance of the results regarding the health of the people.
The article could be published but some revisions are necessary.
1. I consider that the introduction does not provide sufficient background and could to contain more studies linked with the topic of the article.
2. In the final of the Introduction is inserted the purpose of this article. It is said that the purpose is the presentation of the clinical experience of the authors. I think that it is not enough for a scientifically article. It should be highlighted the hypotheses, aim and objectives of the researches.
3. The Statistical analysis should be more detailed and presented in the article.
4. The conclusions must to be correlated with concrete results of the researches.
Author Response
Hi Professor,
Thank you very much for your comments. Your comments are very helpful and we realized the shortcomings of the article. Therefore, we have made corresponding modifications and supplements in our article. Besides, we submitted our manuscript to MDPI for further English editing. The attachment is the edited version.

Round 2
Reviewer 3 Report
The authors have taken in account the recommendations and in this form the article could be published in the journal.